# Bridging the Gap between Synthetic and Authentic Images for Multimodal Machine Translation

**Wenyu Guo**[1,2,3], **Qingkai Fang**[1,2], **Dong Yu**[3], **Yang Feng**[1,2*]

[1]Key Laboratory of Intelligent Information Processing
Institute of Computing Technology, Chinese Academy of Sciences (ICT/CAS)
[2]University of Chinese Academy of Sciences, Beijing, China
[3]Beijing Language and Culture University, China
xk17guowenyu@126.com, yudong@blcu.edu.cn
{fangqingkai21b, fengyang}@ict.ac.cn

## Abstract

Multimodal machine translation (MMT) simultaneously takes the source sentence and a relevant image as input for translation. Since there is no paired image available for the input sentence in most cases, recent studies suggest utilizing powerful text-to-image generation models to provide image inputs. Nevertheless, synthetic images generated by these models often follow different distributions compared to authentic images. Consequently, using authentic images for training and synthetic images for inference can introduce a distribution shift, resulting in performance degradation during inference. To tackle this challenge, in this paper, we feed synthetic and authentic images to the MMT model, respectively. Then we minimize the gap between the synthetic and authentic images by drawing close the input image representations of the Transformer Encoder and the output distributions of the Transformer Decoder. Therefore, we mitigate the distribution disparity introduced by the synthetic images during inference, thereby freeing the authentic images from the inference process. Experimental results show that our approach achieves state-of-the-art performance on the Multi30K En-De and En-Fr datasets, while remaining independent of authentic images during inference.

## 1 Introduction

Multimodal machine translation (MMT) integrates visual information into neural machine translation (NMT) to enhance language understanding with visual context, leading to improvement in translation quality (Li et al., 2022a; Guo et al., 2022; Fang and Feng, 2022). However, most existing MMT models require an associated image with the input sentence, which is difficult to satisfy at inference and limits the usage scenarios. Hence, overcoming

---

*Corresponding author: Yang Feng.
Code is publicly available at https://github.com/ictnlp/SAMMT.

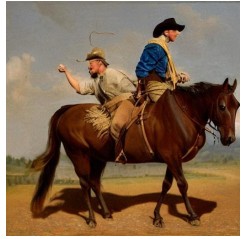 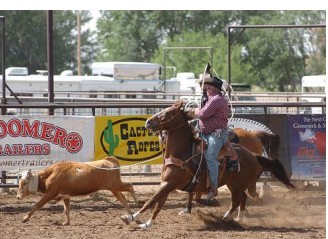

Synthetic                    Authentic
**src**: on horseback a man attempts to rope a young bull

Figure 1: An example of the synthetic and authentic images of the same source sentence.

the dependence on images at inference stands as a pivotal challenge in MMT.

To tackle this challenge, some researchers have proposed to employ text-to-image generation models (Long et al., 2021; Li et al., 2022b) to generate synthetic images to use as the associated image for the source text at inference, while during training MMT models still use the available authentic images as the visual context to generate translation. Despite the remarkable capabilities of text-to-image generation models in generating highly realistic images from textual descriptions (Ramesh et al., 2022; Rombach et al., 2022), the synthetic images may exhibit different distribution patterns compared to the authentic (ground truth) images. As shown in Figure 1, the synthetic images may depict counterfactual scenes, omit information in the text, or add information irrelevant from the text. Therefore, training with authentic images but inferring with synthetic images causing the distribution shift in the images, leading to a decline in translation performance.

In this paper, we embrace the utilization of text-to-image generation models in MMT and propose a method to bridge the gap between synthetic and authentic images. In our method, we also introduce synthetic images during training and feed synthetic images and authentic images to the MMT model, respectively. Then we minimize the gap

between the two images by drawing close the following two terms of the MMT model when the two images are inputted respectively: the input image representations to the Transformer Encoder and the output distributions from the Transformer Decoder. Regarding the input representations, we leverage the Optimal Transport (OT) theory to mitigate the disparity of the representations. Kullback-Leibler (KL) divergence is utilized to ensure the consistency of the output distributions. Consequently, we effectively eliminate the disparities introduced by synthetic images used during the inference process, thereby freeing the authentic images from the inference process.

Experimental results show that our approach achieves state-of-the-art performance on the Multi30K En-De and En-Fr datasets, even in the absence of authentic images. Further analysis demonstrates that our approach effectively enhances the representation and prediction consistency between synthetic and authentic images.

## 2 Background

### 2.1 Stable Diffusion

Stable Diffusion (Rombach et al., 2022) is a text-to-image generation model based on the latent diffusion model (LDM), enabling the generation of highly realistic images from textual descriptions. It mainly consists of a VAE model, a U-Net model and a CLIP text encoder.

The training of the conditional LDM involves a forward diffusion process and a backward denoising process. During the diffusion process, the images are firstly compressed into the latent space using the VAE encoder, followed by the addition of Gaussian noise to the latent representations. In the denoising process, the U-Net is iteratively employed to progressively eliminate noise from the noisy representations. To integrate textual guidance during the denoising process, the text description is encoded using the CLIP text encoder and incorporated into the U-Net through the cross-attention mechanism. Finally, the VAE decoder reconstructs images from the latent representations. By leveraging image-text pairs, the conditional LDM can be optimized by:

$$\mathcal{L}_{ldm} = \mathbb{E}_{\mathcal{E}(a), x, \epsilon \sim \mathcal{N}(0,1), t} \left[ \|\epsilon - \epsilon_\theta(z_t, t, \tau_\theta(x))\|_2^2 \right], \quad (1)$$

where $x$ and $a$ represent the input text and image, respectively. $\epsilon$ denotes the Gaussian noise. $t$ and $z_t$ refer to the time step and the latent representation at the $t$-th time step. $\mathcal{E}$ represents the VAE encoder. $\epsilon_\theta$ and $\tau_\theta$ represent the U-Net and the CLIP text encoder, respectively. In this paper, we utilize the pre-trained Stable Diffusion model to generate images for the source sentence.

### 2.2 Multimodal Transformer

Multimodal Transformer (Yao and Wan, 2020) is a powerful architecture designed for MMT. It replaces the self-attention layer in the Transformer (Vaswani et al., 2017) Encoder with a *multimodal self-attention* layer, which can learn multimodal representations from text representations under the guidance of the image-aware attention.

Formally, given the input text $x$ and image $a$, the textual and visual representations can be denoted as $\mathbf{H}^x = (h_1^x, .., h_N^x)$ and $\mathbf{H}^a = (h_1^a, ..., h_P^a)$, respectively. Here, $N$ represents the length of the source sentence, and $P$ denotes the size of visual features. In each multimodal self-attention layer, the textual and visual representations are concatenated together as the query vectors:

$$\widetilde{\mathbf{H}} = [\mathbf{H}^x; \mathbf{H}^a W^a] \in \mathbb{R}^{(N+P) \times d}, \quad (2)$$

where $W^a$ are learnable weights. The key and value vectors are the textual representations $\mathbf{H}^x$. Finally, the output of the multimodal self-attention layer is calculated as follows:

$$\mathbf{c}_i = \sum_{j=1}^{N} \widetilde{\alpha}_{ij}(h_j^x W^V), \quad (3)$$

where $\widetilde{\alpha}_{ij}$ is the weight coefficient computed by the softmax function:

$$\widetilde{\alpha}_{ij} = \text{softmax}\left( \frac{(\widetilde{h}_i W^Q)(h_j^x W^K)^{\text{T}}}{\sqrt{d_k}} \right), \quad (4)$$

where $W^Q, W^K$, and $W^V$ are learnable weights, and $d_k$ is the dimension of the key vector. The output of the last Multimodal Transformer Encoder layer is fed into the Transformer Decoder to generate the target translation. In this paper, we use the Multimodal Transformer as our base model architecture.

## 3 Method

The original Multimodal Transformer typically requires a paired image of the input sentence to provide visual context, which is often impractical in

many scenarios. To address this limitation, we leverage the Stable Diffusion model during inference to generate a synthetic image based on the source sentence. However, as synthetic images may follow different distributions compared to the authentic (ground truth) images, using authentic images for training and synthetic images for inference could result in a performance decline in translation quality due to the distribution shift in the images. Next, we will introduce our proposed approach for bridging the gap between synthetic and authentic images.

### 3.1 Training with both Synthetic and Authentic Images

To address the issue of distribution shifts in images, we propose training the MMT model using a combination of synthetic and authentic images. Specifically, we enhance the training set by augmenting each sentence with an additional synthetic image, and train the model with both synthetic and authentic images. By incorporating synthetic images during training, we can alleviate the inconsistency between training and inference phases.

Technically, we utilize the Stable Diffusion model to generate an image $s$ corresponding to the input text $x$. We employ the pre-trained CLIP (Radford et al., 2021) image encoder to encode images into visual embeddings. Subsequently, we incorporate a feed-forward network (FFN) to adjust the dimension of visual embeddings to align with the dimension of word embeddings.

$$\mathbf{H}^a = \text{FFN}(\text{CLIPImageEncoder}(a)), \quad (5)$$
$$\mathbf{H}^s = \text{FFN}(\text{CLIPImageEncoder}(s)), \quad (6)$$

where $\mathbf{H}^a \in \mathbb{R}^{1 \times d}$ and $\mathbf{H}^s \in \mathbb{R}^{1 \times d}$ denote visual representations of the authentic and synthetic images, respectively, and $d$ is the dimension of word embeddings. Next, $\mathbf{H}^a$ and $\mathbf{H}^s$ are fed into the Multimodal Transformer to perform the MMT task.

Let us denote the target sentence as $y = (y_1, ..., y_M)$. The loss functions for training with synthetic and authentic images can be formulated as follows:

$$\mathcal{L}_{syn} = -\sum_{j=1}^{M} \log p\left(y_j | y_{<j}, x, s\right), \quad (7)$$

$$\mathcal{L}_{aut} = -\sum_{j=1}^{M} \log p\left(y_j | y_{<j}, x, a\right). \quad (8)$$

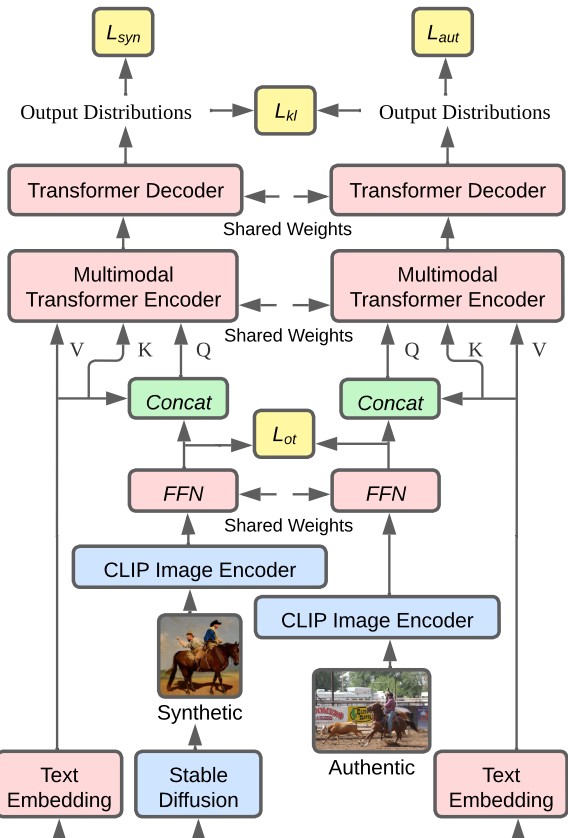

src: on horseback a man attempts to rope a young bull

Figure 2: Overview of our proposed method.

The final training objective is as follows:

$$\mathcal{L}_{trans} = \frac{1}{2}\left(\mathcal{L}_{syn} + \mathcal{L}_{aut}\right). \quad (9)$$

### 3.2 Consistency Training

Incorporating synthetic images into the training process helps mitigate the distribution shift between training and inference. However, the model's behaviour may still differ when handling synthetic and authentic images. To enhance the inherent consistency of the model when dealing with these two types of images, we introduce two consistency losses during training. Optimal transport loss is a common training object used to encourage word-image alignment in vision-language pre-training tasks (Chen et al., 2020; Kim et al., 2021). The Kullback–Leibler divergence loss can also be used to improve visual representation consistency in text-guided image inpainting tasks (Zhou and Long, 2023). Following previous works, at the encoder side, we incorporate an optimal transport-based training objective to encourage consistency between representations of synthetic and authentic

images. At the decoder side, we encourage consistency between the predicted target distributions based on both types of images. Figure 2 illustrates the overview of our proposed method.

**Representation Consistency** Intuitively, we hope the visual representations of synthetic and authentic images to be similar. However, directly optimizing the cosine similarity or L2 distance between $\mathbf{H}^s$ and $\mathbf{H}^a$ may not be optimal, as the visual representations of two images may exhibit different distributions across the feature dimension. Therefore, we propose measuring the representation similarity between synthetic and authentic images based on the optimal transport distance.

The Optimal Transport (OT) problem is a well-known mathematical problem that seeks to find the optimal mapping between two distributions with minimum cost. It has been widely employed as a loss function in various machine learning applications. Following the principles of optimal transport theory (Villani, 2009), the two probability distributions $P$ and $Q$ can be formulated as:

$$
\begin{aligned}
P &= \{(w_i, m_i)\}_{i=1}^{K}, \quad s.t. \sum_i m_i = 1; \\
Q &= \{(\hat{w}_j, \hat{m}_j)\}_{j=1}^{\hat{K}}, \quad s.t. \sum_j \hat{m}_j = 1,
\end{aligned} \tag{10}
$$

where each data point $w_i$ has a probability mass $m_i \in [0, \infty)$. Given a transfer cost function $c(w_i, w_j)$, and using $\mathbf{T}_{ij}$ to represent the mass transported from $w_i$ to $\hat{w}_j$, the transport cost can be defined as:

$$
\begin{aligned}
\mathcal{D}(P, Q) &= \min_{\mathbf{T} \geq 0} \sum_{i,j} \mathbf{T}_{ij} c(w_i, \hat{w}_j), \\
s.t. \quad & \sum_{j=1}^{\hat{K}} \mathbf{T}_{ij} = m_i, \forall i \in \{1, ..., K\}, \\
& \sum_{i=1}^{K} \mathbf{T}_{ij} = \hat{m}_j, \forall j \in \{1, ..., \hat{K}\}.
\end{aligned} \tag{11}
$$

We regard two visual representations $\mathbf{H}^s = (h_1^s, ..., h_d^s)$ and $\mathbf{H}^a = (h_1^a, ..., h_d^a)$ as two independent distributions. Each $h_i^s$ and $h_j^a$ here is a scalar value. We formulate the distance between $\mathbf{H}^s$ and

$\mathbf{H}^a$ as an optimal transport problem, given by:

$$
\begin{aligned}
\mathcal{D}(\mathbf{H}^s, \mathbf{H}^a) &= \min_{\mathbf{T} \geq 0} \sum_{i,j} \mathbf{T}_{ij} c(h_i^s, h_j^a), \\
s.t. \quad & \sum_{j=1}^{d} \mathbf{T}_{ij} = m_i, \forall i \in \{1, ..., d\}, \\
& \sum_{i=1}^{d} \mathbf{T}_{ij} = \hat{m}_j, \forall j \in \{1, ..., d\}.
\end{aligned} \tag{12}
$$

We use the L2 distance as the cost function $c$, and the probability mass is defined as:

$$
\begin{aligned}
m_i &= \frac{|h_i^s|}{\sum_i |h_i^s|}, \\
\hat{m}_j &= \frac{|h_j^a|}{\sum_j |h_j^a|}.
\end{aligned} \tag{13}
$$

Following Kusner et al. (2015), we remove the second constraint to obtain a lower bound of the accurate OT solution. The relaxed OT improves the training speed without performance decline, which can be defined as:

$$
\begin{aligned}
\mathcal{D}(\mathbf{H}^s, \mathbf{H}^a) &= \min_{\mathbf{T} \geq 0} \sum_{i,j} \mathbf{T}_{ij} c(h_i^s, h_j^a), \\
s.t. \quad & \sum_{j=1}^{d} \mathbf{T}_{ij} = m_i, \forall i \in \{1, ..., d\}.
\end{aligned} \tag{14}
$$

The final OT loss is defined as follows:

$$
\mathcal{L}_{ot} = \frac{1}{2} \left( \mathcal{D}(\mathbf{H}^s, \mathbf{H}^a) + \mathcal{D}(\mathbf{H}^a, \mathbf{H}^s) \right), \tag{15}
$$

**Prediction Consistency** In addition to ensuring the consistency of representations between synthetic and authentic images, we also aim to enhance the model's consistency in the predicted probability distributions based on both two types of images. Inspired by previous works on speech translation (Fang et al., 2022; Zhou et al., 2023; Fang and Feng, 2023), we introduce a prediction consistency loss, defined as the Kullback-Leibler (KL) divergence between the two probability distributions:

$$
\mathcal{L}_{kl} = \sum_{j=1}^{M} \text{KL}\left[ p(y_j|y_{<j}, x, s) \| p(y_j|y_{<j}, x, a) \right]. \tag{16}
$$

Finally, the training objective is as follows:

$$
\mathcal{L} = \mathcal{L}_{trans} + \lambda \mathcal{L}_{kl} + \gamma \mathcal{L}_{ot}, \tag{17}
$$

where $\lambda$ and $\gamma$ are the hyperparameters that control the contribution of the KL divergence loss and the optimal transport loss.

| # | Models | En-De | | | En-Fr | | | Average |
|---|--------|-------|---|---|-------|---|---|---------|
| | | Test2016 | Test2017 | MSCOCO | Test2016 | Test2017 | MSCOCO | |
| | *Text-only Transformer* | | | | | | | |
| 1 | TEXT-ONLY | 40.69 | 34.26 | 30.52 | 62.84 | 54.35 | 44.81 | 44.58 |
| | *Previous Image-must Systems* | | | | | | | |
| 2 | [†]DCCN (Lin et al., 2020) | 39.70 | 31.00 | 26.70 | 61.20 | 54.30 | 45.40 | 43.05 |
| 3 | [†]RMMT (Wu et al., 2021) | 41.45 | 32.94 | 30.01 | 62.12 | 54.39 | 44.52 | 44.24 |
| 4 | [†]Doubly-ATT (Calixto et al., 2017) | 41.45 | 33.95 | 29.63 | 61.99 | 53.72 | 45.16 | 44.32 |
| 5 | [†]Gated Fusion (Wu et al., 2021) | 41.96 | 33.59 | 29.04 | 61.69 | 54.85 | 44.86 | 44.33 |
| 6 | Selective Attention (Li et al., 2022a) | 41.84 | 34.32 | 30.22 | 62.24 | 54.52 | 44.82 | 44.66 |
| 7 | Noise-robust (Ye et al., 2022) | 42.56 | 35.09 | 31.09 | 63.24 | 55.48 | 46.34 | 45.63 |
| 8 | VALHALLA (Li et al., 2022b) | **42.60** | 35.10 | 30.70 | 63.10 | 56.00 | 46.40 | 45.65 |
| | *Our Image-must Systems* | | | | | | | |
| 9 | MULTITRANS (A) (Yao and Wan, 2020) | 41.46 | 34.36 | 30.07 | 62.57 | 54.92 | 44.53 | 44.65 |
| 10 | INTEGRATED (A) | 42.21 | 33.94 | 31.05 | 62.76 | 54.73 | 45.18 | 44.98 |
| 11 | OURS (A) | 42.50 | **36.04** | **31.95** | **63.71** | **56.17** | **46.43** | **46.13** |
| | *Previous Image-free Systems* | | | | | | | |
| 12 | [†]UVR-NMT (Zhang et al., 2020) | 40.79 | 32.16 | 29.02 | 61.00 | 53.20 | 43.71 | 43.31 |
| 13 | [†]Imagination (Elliott and Kádár, 2017) | 41.31 | 32.89 | 29.90 | 61.90 | 54.07 | 44.81 | 44.15 |
| 14 | Distill (Peng et al., 2022) | 41.28 | 33.83 | 30.17 | 62.53 | 54.84 | - | - |
| 15 | VALHALLA (Li et al., 2022b) | **42.70** | 35.10 | 30.70 | 63.10 | 56.00 | **46.50** | 45.68 |
| | *Our Image-free Systems* | | | | | | | |
| 16 | MULTITRANS (S) (Yao and Wan, 2020) | 41.46 | 34.36 | 30.07 | 62.54 | 54.82 | 44.39 | 44.61 |
| 17 | INTEGRATED (S) | 42.21 | 33.94 | 31.05 | 62.76 | 54.73 | 45.14 | 44.97 |
| 18 | OURS (S) | 42.50 | **36.04** | **31.95** | **63.71** | **56.17** | 46.43 | **46.13** |

Table 1: BLEU scores on Multi30K Test2016, Test2017 and MSCOCO test sets of En-De and En-Fr tasks. "(A)" indicates using authentic images during the inference stage, while "(S)" indicates using synthetic images. [†] indicates results under the Transformer-Tiny configuration, which are quoted from Wu et al. (2021).

## 4 Experiments

### 4.1 Dataset

We conduct experiments on the Multi30K dataset (Elliott et al., 2016). Multi30K contains bilingual parallel sentence pairs with image annotations, where the English description of each image is manually translated into German (De) and French (Fr). The training and validation sets consist of 29,000 and 1,014 instances, respectively. We report the results on the Test2016, Test2017 and ambiguous MSCOCO test sets (Elliott et al., 2017), which contain 1,000, 1,000 and 461 instances respectively. We apply the byte-pair encoding (BPE) (Sennrich et al., 2016) algorithm with 10K merge operations to segment words into subwords, resulting in a shared vocabulary of 9,712 and 9,544 entries for the En-De and En-Fr translation tasks, respectively.

### 4.2 System Settings

**Stable Diffusion** We build our own inference pipeline with diffusers (von Platen et al., 2022). We use the pre-trained VAE and U-Net models[1].

The pre-trained CLIP (Radford et al., 2021) text encoder[2] is used to encode the input text. The number of denoising steps is set to 50. The seed of the generator is set to 0. The scale of classifier-free guidance is 7.5 and the batch size is 1.

**Visual Features** For both synthetic and authentic images, we use the pre-trained ViT-B/32 CLIP model[3] to extract the visual features, which have a dimension of $[1 \times 512]$.

**Translation Model** We follow Wu et al. (2021) to conduct experiments with the Transformer-Tiny configuration, which proves to be more effective on the small dataset like Multi30K. The translation model consists of 4 encoder and decoder layers. The hidden size is 128 and the filter size of FFN is 256. There are 4 heads in the multi-head self-attention module. The dropout is set to 0.3 and the label smoothing is 0.1. Our implementation is based on the open-source framework *fairseq*[4] (Ott et al., 2019). Each training batch contains 2,048 tokens and the update frequency is set to 4.

[1] https://huggingface.co/CompVis/stable-diffusion-v1-4

[2] https://huggingface.co/openai/clip-vit-large-patch14

[3] https://github.com/openai/CLIP

[4] https://github.com/facebookresearch/fairseq

| # | $\mathcal{L}_{kl}$ | $\mathcal{L}_{ot}$ | En-De | | | Average |
|---|---|---|---|---|---|---|
| | | | Test2016 | Test2017 | MSCOCO | |
| 1 | × | × | 42.21 | 33.94 | 31.05 | 35.73 |
| 2 | × | ✓ | 41.71 | 33.57 | 29.42 | 34.90 |
| 3 | ✓ | × | 42.31 | 35.76 | 31.04 | 36.37 |
| 4 | ✓ | ✓ | **42.50** | **36.04** | **31.95** | **36.83** |

Table 2: BLEU scores on three test sets with different training objectives in the `image-free` setting.

| Model | Similarity |
|---|---|
| **MULTITRANS** | 39.43% |
| **INTEGRATED** | 86.60% |
| **OURS** (S) | 100.00% |

Table 3: The average cosine similarity between visual representations of the synthetic and authentic images on the Test2016 test set.

For evaluation, we average the last 10 checkpoints following previous works. The beam size is set to 5. We measure the results with BLEU (Papineni et al., 2002) scores for all test sets. All models are trained and evaluated using 1 Tesla V100 GPU. We set the KL loss weight $\lambda$ to 0.5. The OT loss weight $\gamma$ is set to 0.1 for En-De and 0.9 for En-Fr.

### 4.3 Basline Systems

We conduct experiments in two different settings: `image-must` and `image-free`. In the `image-must` setting, authentic images from the test set are used to provide the visual context. In contrast, the authentic images are not used in the `image-free` setting. We implement the following three baseline systems for comparision:

**TEXT-ONLY** Text-only Transformer is implemented under the Transformer-Tiny configuration. It follows an encoder-decoder paradigm (Vaswani et al., 2017), taking only texts as input.

**MULTITRANS** Multimodal Transformer (Yao and Wan, 2020) trained on the original Multi30K dataset, as described in Section 2.2.

**INTEGRATED** Multimodal Transformer trained on our augmented Multi30K dataset, where each sentence is paired with an authentic image and a synthetic image, as described in Section 3.1.

Besides, in the `image-must` setting, we include DCCN (Lin et al., 2020), RMMT (Wu et al., 2021), Doubly-ATT (Calixto et al., 2017), Gated Fusion (Wu et al., 2021), Selective Attention (Li et al., 2022a), Noise-robust (Ye et al., 2022), and VAL-HALLA (Li et al., 2022b) for comparison. In the `image-free` setting, we include UVR-NMT (Zhang et al., 2020), Imagination (Elliott and Kádár, 2017), Distill (Peng et al., 2022), and VAL-HALLA (Li et al., 2022b) for comparison.

### 4.4 Main Results on the Multi30K Dataset

Table 1 summarizes the results in both `image-must` and `image-free` settings. Each model is evaluated on three test sets for two language pairs.

Firstly, our method and the majority of the baseline systems demonstrate a substantial performance advantage over the **TEXT-ONLY** baseline, underscoring the importance of the visual modality.

Secondly, for the **MULTITRANS** baseline, it is evident that the synthetic and authentic images yield different translation results. Overall, the authentic images outperform the synthetic images. Directly substituting authentic images with synthetic images results in a performance decline due to the distribution shift in images.

Thirdly, for our **INTEGRATED** baseline, we incorporate synthetic images into the training process. Experimental results show some improvements compared to the **MULTITRANS** baseline. The utilization of synthetic or authentic images during inference does not significantly affect the results, indicating the effectiveness of this approach in addressing the distribution shift between training and inference.

Finally, our method significantly outperforms all baseline systems in both `image-must` and `image-free` settings. It achieves state-of-the-art performance on all test sets, while remaining independent of the authentic images. The superior performance demonstrates the effectiveness of encouraging the representation and prediction consistency between synthetic and authentic images in the MMT training.

## 5 Analysis

### 5.1 Effects of Training Objectives

To investigate the impact of the KL divergence (KL) loss and the optimal transport (OT) loss, we conduct ablation experiments on the training objectives. The results are presented in Table 2. When only applying the OT loss (#2), we observe a performance decline. When only using the KL loss (#3), we observe an improvement of 0.64 BLEU, highlighting the benefits of promoting prediction consistency. When both KL loss and OT loss are utilized (#4), we achieve a more significant improvement

| Model | En-De | | | Average |
|---|---|---|---|---|
| | Test2016 | Test2017 | MSCOCO | |
| **SELECTIVE** (A) | 41.64 | 33.88 | 29.92 | 35.15 |
| **INTEGRATED** (A) | 41.21 | 34.31 | 30.10 | 35.21 |
| **OURS** (A) | **42.42** | **35.44** | **31.88** | **36.58** |
| **SELECTIVE** (S) | 41.64 | 33.88 | 29.92 | 35.15 |
| **INTEGRATED** (S) | 41.21 | 34.31 | 30.10 | 35.21 |
| **OURS** (S) | **42.42** | **35.44** | **31.88** | **36.58** |

Table 4: BLEU scores of three systems built upon the Selective Attention (Li et al., 2022a) model.

| Model | En-De | | | Average |
|---|---|---|---|---|
| | Test2016 | Test2017 | MSCOCO | |
| **OURS** (S) | **42.50** | **36.04** | **31.95** | **36.83** |
| **RANDOM** | 41.96 | 35.68 | 31.20 | 36.28 |
| **NOISE** | 41.86 | 35.72 | 30.77 | 36.12 |

Table 5: BLEU scores of our approach and other two regularization methods.

| Model | En-De | | | Average |
|---|---|---|---|---|
| | Test2016 | Test2017 | MSCOCO | |
| **OURS** (S) | **42.50** | **36.04** | 31.95 | **36.83** |
| **COSINE** | 42.46 | 35.53 | **32.10** | 36.70 |
| L2 | 42.48 | 36.00 | 31.77 | 36.75 |

Table 6: BLEU scores of our approach and other two loss functions.

of 1.10 BLEU, demonstrating the benefits of simultaneously encouraging the representation and prediction consistency.

## 5.2 Representation Consistency

To investigate the effectiveness of our method in aligning the representations of synthetic and authentic images, we measure the cosine similarity between their respective representations ($\mathbf{H}^s$ and $\mathbf{H}^a$) on the test set. Specifically, we measure the cosine similarity of the visual representations after the shared-weights FFN network on Test2016 of En-De translation task during inference. Table 3 presents the results. For the **MULTITRANS** baseline, a significant disparity exists between the visual representations of synthetic and authentic images. However, when the synthetic images are integrated into the training process (**INTEGRATED**), the similarity between synthetic and authentic images increases to 86.60%. In contrast, our method achieves an impressive similarity of 100.00%, highlighting the effectiveness of our approach in bridging the representation gap between synthetic and authentic images. The optimal transport loss plays a crucial role in reducing the representation gap.

## 5.3 Apply Our Method to Other Model Architecture

Our approach focuses on improving the training methodology for MMT, which is not restricted to a specific model architecture. To validate the generality of our method, we conduct experiments using an alternative model architecture: Selective Attention

(**SELECTIVE**) (Li et al., 2022a), which incorporating visual representations with a selective attention module and a gated fusion module. We apply the optimal transport loss before the selective attention module, and apply the KL loss in the decoder. The model settings and hyperparameters are the same as our experiments on the Multimodal Transformer.

The results are shown in Table 4. For the **IN-TEGRATED** baseline, integrating synthetic images into the training process can eliminate the gap and bring slight improvements. When we add the KL loss and OT loss during training, we observe a 1.43 BLEU improvements compared with the **SE-LECTIVE** baseline, demonstrating the effectiveness and generalizability of our method across different model architectures.

## 5.4 Comparison with Regularization Methods

We employ the Stable Diffusion model to generate images from textual descriptions as visual contexts. To verify the necessity of the text-to-image generation model, we compare its performance with other two regularization methods: **RANDOM** and **NOISE**. **RANDOM** means shuffling the correspondences between synthetic images and textual descriptions in the training set. **NOISE** means using noise vectors as the visual representations. Results in Table 5 show that both two strategies perform worse than our approach, demonstrating that our superior performance is not solely attributed to regularization effects but rather to the utilization of appropriate images and semantic information.

## 5.5 Comparison with Other Loss Functions

We measure the representation similarity between synthetic and authentic images, and minimize their distance during training. To assess the significance of the optimal transport loss, we conduct experiments using various loss functions. Table 6 presents the results when substituting the optimal transport loss with the **COSINE** embedding loss and L2 loss functions in order to improve representation consistency. The hyperparameters are the

| Model | En-Cs | | Average | MSCTD |
|---|---|---|---|---|
| | Test2016 | Test2018 | | |
| TEXT-ONLY | 34.01 | 29.46 | 31.73 | 21.37 |
| MULTITRANS (S) | 34.18 | 30.03 | 32.11 | 21.48 |
| INTEGRATED (S) | 33.50 | 29.62 | 31.56 | 21.65 |
| OURS (S) | **35.23** | **31.19** | **33.21** | **24.40** |

Table 7: BLEU scores of our approach on different languages and datasets.

| Model | ΔBLEU |
|---|---|
| **MULTITRANS(S)** | 0.23 |
| **INTEGRATED(S)** | 0.43 |
| **OURS(S)** | **0.56** |

Table 8: Results of the Incongruent Decoding on the Multi30K En-De test sets.

same as our experiments on the optimal transport loss. The drop of results indicates the effectiveness of the optimal transport loss in mitigating the representation disparity.

## 5.6 Results on Different Languages and Datasets

To verify that our method can be utilized in different language pairs and datasets, we conduct experiments on the En-Cs task of the Multi30K dataset and the En-De translation task of the MSCTD dataset (Liang et al., 2022). The MSCTD dataset is a multimodal chat translation dataset, including 20,240, 5,063 and 5,047 instances for the training, validation and test sets respectively. The results are shown in Table 7. Our method still achieves significant improvements among all the baselines on the En-Cs task and the MSCTD dataset, showing the generality of our method among different languages and datasets.

## 5.7 Incongruent Decoding

We follow method (Elliott, 2018) to use an adversarial evaluation method to test if our method is more sensitive to the visual context. We set the congruent image's feature to all zeros. Then we observe the value ΔBLEU by calculating the difference between the congruent data and the incongruent one. A larger value means the model is more sensitive to the image context. As shown in Table 8, we conduct experiment on the three test sets of the En-De translation task and calculate the average incongruent results. Notably, our method achieves the highest score, providing substantial evidence of its sensitivity to visual information.

## 5.8 Case Study

Table 9 presents two translation cases in the image-free scenario involving the four systems. In the first case, our model accurately translates the phrase "im maul" (in the mouth) and "rennt im schnee" (run in the snow), whereas the other systems provide incomplete translations. In the second

case, the phrases "spielt" (play) and "auf einer roten gitarre" (on a red guitar) are correctly translated by our proposed model while other systems translate inaccurately. To verify that our system benefits from the incorporation of visual information, we also compute the BLEU score for each sentence in the En-De Test2016 test set and conduct a comparison between our model and the TEXT-ONLY baseline. Among 1,000 data instances, 34% achieve higher blue scores upon incorporating visual information, whereas 24.9% exhibit lower blue scores. We also manually annotated the En-De Test2016 test set, enlisting a German language expert who is certified at the professional eighth level for the annotation task. The annotation results reveal that 35.7% of the instances benefit from the visual features, while 30.8% show no improvement from visual features. These results demonstrate that our model enhances translation performance when visual information is incorporated, further confirming the improvement in translation results achieved by aligning synthetic and authentic images.

## 6 Related Work

### 6.1 Multimodal Machine Translation

Multimodal Machine translation (MMT) has drawn great attention in the research community. Existing methods mainly focus on integrating visual information into Neural Machine Translation (NMT) (Zhang et al., 2023). Multimodal Transformer (Yao and Wan, 2020) employs a graph perspective to fuse visual information into the translation model. Gated and RMMT, introduced by Wu et al. (2021), are two widely-used architectures for integrating visual and textual modalities. Selective Attention (Li et al., 2022a) is a method that utilizes texts to select useful visual features. Additionally, Ji et al. (2022) proposes additional training objectives to enhance the visual awareness of the MMT model. Yang et al. (2022) introduces cross-modal contrastive learning to enhance low-resource NMT.

In addition to methods that rely on authentic images, there is a growing interest in developing

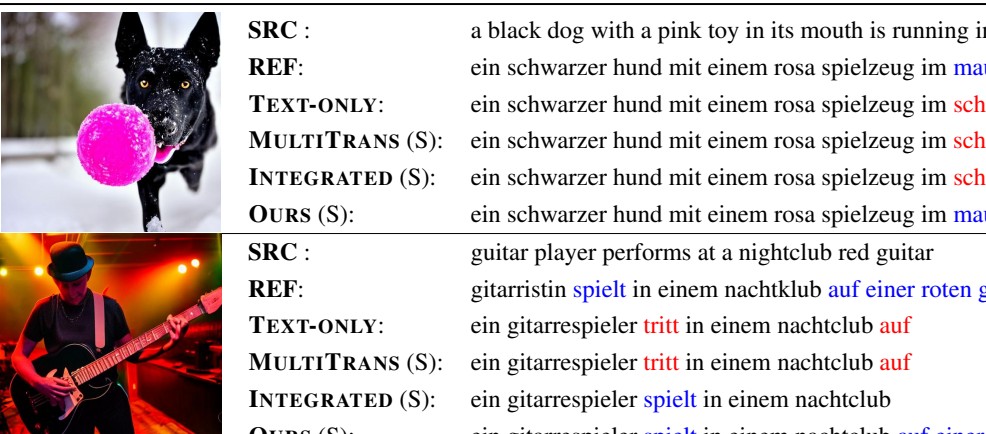

| | | |
|---|---|---|
| **SRC** : | | a black dog with a pink toy in its mouth is running in the snow |
| **REF**: | | ein schwarzer hund mit einem rosa spielzeug im maul rennt im schnee |
| **TEXT-ONLY**: | | ein schwarzer hund mit einem rosa spielzeug im schnee rennt |
| **MULTITRANS** (S): | | ein schwarzer hund mit einem rosa spielzeug im schnee |
| **INTEGRATED** (S): | | ein schwarzer hund mit einem rosa spielzeug im schnee läuft |
| **OURS** (S): | | ein schwarzer hund mit einem rosa spielzeug im maul rennt im schnee |
| **SRC** : | | guitar player performs at a nightclub red guitar |
| **REF**: | | gitarristin spielt in einem nachtklub auf einer roten gitarre |
| **TEXT-ONLY**: | | ein gitarrespieler tritt in einem nachtclub auf |
| **MULTITRANS** (S): | | ein gitarrespieler tritt in einem nachtclub auf |
| **INTEGRATED** (S): | | ein gitarrespieler spielt in einem nachtclub |
| **OURS** (S): | | ein gitarrespieler spielt in einem nachtclub auf einer roten gitarre |

Table 9: Two translation cases of four systems on the En-De task in the `image-free` scenario. The red and blue tokens denote error and correct translations respectively.

MMT systems in `image-free` scenarios due to the difficulty of obtaining paired images during inference. The image retrieval-based approaches (Zhang et al., 2020; Fang and Feng, 2022) utilize images retrieved based on textual information as visual representations. The proposed method (Li et al., 2022b) generates discrete visual representations from texts and incorporate them in the training and inference process. Peng et al. (2022) proposes distillation techniques to transfer knowledge from image representations to text representations, enabling the acquisition of useful multimodal features during inference. In this paper, we focus on using synthetic visual representations generated from texts and aim to bridge the gap between synthetic and authentic images.

## 6.2 Text-to-image Generation

Text-to-image generation models have made significant progress in generating highly realistic images from textual descriptions. Early approaches (Karras et al., 2018; Xu et al., 2018) rely on training Generative Adversarial Networks (GANs) (Goodfellow et al., 2014) to generate images. CLIP (Radford et al., 2021), a powerful vision-and-language model, has also been utilized to guide the image generation process in various methods (Ramesh et al., 2022; Abdal et al., 2022). Recently, researchers have achieved impressive results in zero-shot text-to-image generation scenarios, where images are generated based on textual descriptions without specific training data (Ramesh et al., 2021; Yu et al., 2022). The adoption of diffusion-based methods (Nichol et al., 2022; Saharia et al., 2022) has further pushed the boundaries of text-to-image

generation, resulting in high-quality image synthesis. In this work, we employ the Stable Diffusion (Rombach et al., 2022) model, a diffusion-based approach, as our text-to-image generation model to generate content-rich images from texts.

## 7  Conclusion

In this paper, we strive to address the disparity between synthetic and authentic images in multimodal machine translation. Firstly, we feed synthetic and authentic images to the MMT model, respectively. Secondly, we minimize the gap between the synthetic and authentic images by drawing close the input image representations of the Transformer Encoder and the output distributions of the Transformer Decoder through two loss functions. As a result, we mitigate the distribution disparity introduced by the synthetic images during inference, thereby freeing the authentic images from the inference process. Through extensive experiments, we demonstrate the effectiveness of our proposed method in improving the quality and coherence of the translation results.

## Limitations

The utilization of the text-to-image generation process may introduce additional computational overhead. Moreover, our method is currently limited to translation tasks from English to other languages, as the pre-trained text-to-image model lacks support for languages other than English. The application of our method to other tasks and languages remains unexplored, and we consider these limitations as potential areas for future research.

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
