# OpenReview forum: "Bridging the Gap between Synthetic and Authentic Images for Multimodal Machine Translation"
_EMNLP/2023/Conference — EMNLP 2023 Main_

### Official Review · Reviewer_ZYWp · 2023-08-03

**Soundness:** 4

**Excitement:**

4: Strong: This paper deepens the understanding of some phenomenon or lowers the barriers to an existing research direction.

**Paper Topic And Main Contributions:**

The paper proposes a method to bridge the gap between synthetic and authentic images for multimodal machine translation (MMT). The proposed method feeds synthetic and authentic images to the MMT model during training and inference, respectively, and minimizes the gap between the two types of images by drawing close the input image representations of the Transformer Encoder and the output distributions of the Transformer Decoder.
The main contributions of the paper are:
- Introducing synthetic images during training to alleviate the inconsistency between training and inference phases.
- Proposing an optimal transport-based training objective to encourage consistency between representations of synthetic and authentic images.
 - Introducing a prediction consistency loss to enhance the model's consistency in the predicted probability distributions based on both types of images.
- Introducing synthetic images during training to alleviate the inconsistency between the training and inference phases.
- Proposing an optimal transport-based training objective to encourage consistency between representations of synthetic and authentic images.
- Introducing a prediction consistency loss to enhance the model's consistency in the predicted probability distributions based on both types of images.

Three technical backgrounds and concepts required to understand the paper are:
 1. Multimodal machine translation: The paper focuses on multimodal machine translation, which requires an understanding of both natural language processing and computer vision.
2. Transformer model: The proposed method utilizes the Transformer model, which is a neural network architecture commonly used in natural language processing tasks.
3. Optimal transport theory: The paper employs the Optimal Transport theory to mitigate the disparity of the representations between synthetic and authentic images. An understanding of this theory is required to comprehend the proposed approach.

**Reasons To Accept:**

 - The paper proposes a new method to bridge the gap between synthetic and authentic images for multimodal machine translation. The proposed method feeds synthetic and authentic images to the MMT model, respectively, and minimizes the gap between the two images by drawing close the input image representations of the Transformer Encoder and the output distributions of the Transformer Decoder.
- The paper provides a comprehensive evaluation against various baseline models and different datasets. While the main dataset is the Multi30K dataset, utilizing other datasets has increased the validity of the evaluation and ablation study.

**Reasons To Reject:**

N/A




**Reproducibility:**

3: Could reproduce the results with some difficulty. The settings of parameters are underspecified or subjectively determined; the training/evaluation data are not widely available.

**Reviewer Confidence:**

2: Willing to defend my evaluation, but it is fairly likely that I missed some details, didn't understand some central points, or can't be sure about the novelty of the work.

---

> ### Author Rebuttal · Authors · 2023-08-28
>
> Thank you for your insightful comments. We conduct experiments utilizing the MSCTD En-De dataset [2] to prove the validity of our proposed method. The training, validation and test sets consist of 20,240, 5,063 and 5,047 instances, respectively. The results are shown in Table 4:
> |**Model** |**BLEU** |
> | --- | :---: |
> | **TEXT-ONLY** | 21.37 |
> | **MULTITRANS(S)** | 21.48 |
> | **INTEGRATED(S)** | 21.65|
> | **OURS(S)** | **24.40** |
>
> Table 4: BLEU scores on the MSCTD En-De test sets.
>
> Our model significantly outperforms the baseline models, demonstrating the generality of our approach across different datasets. We will add these results in the next version.
>
> [2] MSCTD: A Multimodal Sentiment Chat Translation Dataset. ACL 2022.

---

### Official Review · Reviewer_Ft2o · 2023-08-04

**Soundness:** 3

**Excitement:**

3: Ambivalent: It has merits (e.g., it reports state-of-the-art results, the idea is nice), but there are key weaknesses (e.g., it describes incremental work), and it can significantly benefit from another round of revision. However, I won't object to accepting it if my co-reviewers champion it.

**Paper Topic And Main Contributions:**

This paper addresses the challenge of multimodal machine translation (MMT) by proposing a method to bridge the gap between synthetic and authentic images used during the inference process. The main contribution of the paper is the introduction of a technique that feeds both synthetic and authentic images to the MMT model, minimizing the disparity between their representations. This is achieved through the use of Optimal Transport (OT) theory to mitigate differences in input image representations and Kullback-Leibler (KL) divergence to ensure output distribution consistency. By doing so, the paper effectively eliminates the distribution shift caused by using synthetic images during inference, thereby freeing the authentic images from the inference process. The experimental results demonstrate state-of-the-art performance on the Multi30K En-De and En-Fr datasets, even without the need for authentic images.



**Questions For The Authors:**

1. Why does the average cosine similarity between visual representations of the synthetic and authentic images on the Test2016 test set reach 100%? It seems somewhat unbelievable, and I hope the authors can provide further explanation.

2. Although the proposed method achieves state-of-the-art performance on multiple benchmarks, I have always been skeptical about the effectiveness of visual information in improving the translation of the source sentence. I believe that the MMT task lacks an effective evaluation metric for assessing the quality of translations based on the incorporation of image information.

3. Using a single case to illustrate the improvement brought by visual information to the translation may not be sufficient. I recommend the authors to conduct a manual evaluation on a test set, for instance, to determine how many translations benefit from incorporating images and how many translations do not benefit from using visual information. This would provide a more comprehensive understanding of the impact of visual information on the quality of translations.

**Reasons To Accept:**

The paper's strengths lie in its innovative approach to address the dependence on authentic images in multimodal machine translation. The proposed technique for minimizing the gap between synthetic and authentic images is novel and well-founded on principles like Optimal Transport and Kullback-Leibler divergence. The empirical results show significant performance improvements, making it relevant and impactful for the NLP community. The paper's contributions open up new possibilities for using text-to-image generation models in MMT, making it valuable for researchers and practitioners in the field.

**Reasons To Reject:**

One potential weakness of the paper could be its heavy reliance on synthetic images during the training process. Although the proposed method effectively mitigates the distribution shift during inference, the extent of the dependence on synthetic images could raise concerns about the robustness of the approach. Additionally, the method's performance may be sensitive to the quality and diversity of the synthetic images generated. There might also be a risk of overfitting to specific synthetic image characteristics, limiting the generalizability of the proposed technique.

**Reproducibility:**

4: Could mostly reproduce the results, but there may be some variation because of sample variance or minor variations in their interpretation of the protocol or method.

**Reviewer Confidence:**

4: Quite sure. I tried to check the important points carefully. It's unlikely, though conceivable, that I missed something that should affect my ratings.

---

> ### Author Rebuttal · Authors · 2023-08-28
>
> Thank you for your precious advice.
>
> **Q: About the concern of the model's sensitivity of the quality and diversity of the synthetic images.**
>
> A: 1. The reliability of our method on the synthetic images is constrained, because we use the loss functions to enhance the consistency between synthetic and authentic visual representations during the training process.
>
> 2. By configuring the generator parameters denoising step and manual seed of the Stable Diffusion model, the consistency of images generated from the same textual input is ensured.
>
> 3. To validate the robustness of our approach, we conduct experiments using synthetic images generated by the Stable Diffusion model, varying in quality and diversity. We alter the denoising steps and manual seed settings to control these factors. The corresponding outcomes are presented in Table 1. Remarkably, with synthetic images of varying quality and diversity, our model consistently outperforms the baseline systems. Notably, shuffling the correspondences between synthetic images and textual descriptions in the training set, as shown in section 5.4, has minimal impact on translation quality reduction. We also measure the variance of the different results. These findings demonstrate that our method's efficacy is not contingent on the quality and diversity of the synthetic images.
>
> |**Model**  |**Test2016**  |**Test2017**  |**Test2018**  |**Average**|
> | :---: | :---: | :---: | :---: | :---: |
> |**OURS(S)**  |42.50  |36.04  |31.95  |**36.83** |
> |Denoising Step 40  | 42.24 | 35.68 | 32.17 |36.70 |
> |Denoising Step 60 | 42.37 | **36.05** | 31.80 |36.74 |
> |Manual Seed 1 | **42.74** | 34.94 |32.06  | 36.58 |
> |Manual Seed 2 |42.41 |35.41 |**32.41** |36.74 |
> |Variance |0.045 |0.218 |0.064 |0.006 |
>
> Table 1: BLEU scores of varying quality and diversity synthetic images upon En-De test sets.
>
> **Q: Why does the average cosine similarity between visual representations of the synthetic and authentic images on the Test2016 test set reach 100%? It seems somewhat unbelievable, and I hope the authors can provide further explanation.**
>
> A: We measure the cosine similarity of the visual representation after the shared weights FFN network during inference on Test 2016 of En-De translation task. We utilize global visual features instead of local visual features, concentrating more on the overall information of the images and are relatively easy to align. The results of the image-must and image-free system in our method are also the same. With the help of Optimal Transport Loss, we believe that it is feasible to completely eliminate disparities in visual representations.
>
> **Q: Although the proposed method achieves state-of-the-art performance on multiple benchmarks, I have always been skeptical about the effectiveness of visual information in improving the translation of the source sentence. I believe that the MMT task lacks an effective evaluation metric for assessing the quality of translations based on the incorporation of image information.**
>
> A: 1. Compared to the text-only Transformer baseline, the enhancements observed in our proposed method reflect the effectiveness of visual information. The resutls of the text-only neural machine translation baseline are shown in Table 2. These results will be added in the next version of our paper.
>
> |**Model** | **En-De Test2016** |**En-De Test2017** |**En-De MSCOCO**|**En-Fr Test2016** |**En-Fr Test2017** |**En-Fr MSCOCO** |**Average**|
> | --- | --- | --- | --- | --- | --- | --- |--- |
> |**TEXT-ONLY**|40.69 | 34.26 | 30.52 | 62.84 | 54.35 | 44.81 | 44.58 |
>
> Table 2: BLEU scores of the Text-only baseline model.
>
> 2. We follow method [1] to use an adversarial evaluation method to test if our method is more sensitive to the visual context. We set the congruent image's feature to all zeros. Then we observe the value ΔBLEU by calculating the difference between the congruent data and the incongruent one. A larger value means the model is more sensitive to the image context. As shown in Table 3, we conduct experiment on the three test sets of the En-De translation task and calculate the average incongruent result on three test sets. The result of OURS(S) is the highest, which proves our method's sensitivity to visual information.
>
> |**Model**  |**ΔBLEU**  |
> | --- | :---: |
> | **MULTITRANS(S)** | 0.23 |
> | **INTEGRATED(S)** | 0.43 |
> | **OURS(S)** | 0.56 |
>
> Table 3: Results of the Incongruent Decoding upon En-De test sets.
>
> [1] Adversarial evaluation of multimodal machine translation. EMNLP 2018.
>
> **Q: Using a single case to illustrate the improvement brought by visual information to the translation may not be sufficient. I recommend the authors to conduct a manual evaluation on a test set, for instance, to determine how many translations benefit from incorporating images and how many translations do not benefit from using visual information. This would provide a more comprehensive understanding of the impact of visual information on the quality of translations.**
>
> A: We compute the BLEU score for each sentence in the En-De Test2016 test set and conduct a comparison between our model and the text-only baseline. Among 1000 data instances, 34% show improvement upon incorporating visual information, whereas 24.9% do not exhibit improvement from the utilization of visual information.
> We also manually annotated the En-De Test2016 test set, enlisting a German language expert who is certified at the professional eighth level for the annotation task. By scoring both the text-only translation system and our translation system, we investigate whether the inclusion of visual information indeed enhances translation quality. The annotation results reveal that among 1000 data instances, 35.7% benefit from the utilization of visual features, while 30.8% show no improvement from visual features. Through the BLEU score and manual annotation, it becomes evident that the number of translation results benefiting from the use of visual features is greater than those not benefiting from them, signifying the capacity of visual features to enhance translation quality within our translation system.

---

### Official Review · Reviewer_JtfV · 2023-08-05

**Soundness:** 3

**Excitement:**

3: Ambivalent: It has merits (e.g., it reports state-of-the-art results, the idea is nice), but there are key weaknesses (e.g., it describes incremental work), and it can significantly benefit from another round of revision. However, I won't object to accepting it if my co-reviewers champion it.

**Missing References:**

[1] UNITER: UNiversal Image-TExt Representation Learning. ECCV 2020.\
[2] ViLT: Vision-and-Language Transformer Without Convolution or Region Supervision. ICML 2021.\
[3] Improving Cross-modal Alignment for Text-Guided Image Inpainting. EACL 2023.

**Paper Topic And Main Contributions:**

This paper proposes to train the multimodal machine translation model with both synthetic and authentic images to mitigate the distribution shift caused by using different types of images for training and inference. To enhance the inherent consistency of the model when handling synthetic and authentic images, the paper introduces two consistency losses that align their representations at the encoder side and their predictions at the decoder side.

**Questions For The Authors:**

As mentioned in "Reasons To Reject" above

**Reasons To Accept:**

- The work proposes to reduce the distribution discrepancy between the training and inference phases by introducing two consistency losses.
- The writing and organization of the paper are well done, making it easy to follow and understand.

**Reasons To Reject:**

- The method proposed in the paper is a commonly used objective in visual-text models[1,2], and is not a new approach.
- The Representation Consistency and Prediction Consistency proposed in the paper has already been used in other methods[3].

[1] UNITER: UNiversal Image-TExt Representation Learning. ECCV 2020.\
[2] ViLT: Vision-and-Language Transformer Without Convolution or Region Supervision. ICML 2021.\
[3] Improving Cross-modal Alignment for Text-Guided Image Inpainting . EACL 2023.

**Reproducibility:**

2: Would be hard pressed to reproduce the results. The contribution depends on data that are simply not available outside the author's institution or consortium; not enough details are provided.

**Reviewer Confidence:**

4: Quite sure. I tried to check the important points carefully. It's unlikely, though conceivable, that I missed something that should affect my ratings.

---

> ### Author Rebuttal · Authors · 2023-08-28
>
> **Q: The method proposed in the paper is a commonly used objective in visual-text models[1,2], and is not a new approach.
> The Representation Consistency and Prediction Consistency proposed in the paper has already been used in other methods[3].**
>
> A: Thank you for your invaluable insights. While loss functions have previously been employed to generate multimodal representations, we have pioneered the application of these techniques in the realm of multimodal machine translation. We find the results indeed intriguing and these techniques highly beneficial for leveraging the synthetic images in multimodal machine translation. Furthermore, these techniques hold the potential to alleviate the reliance on authentic images during the inference phase in multimodal machine translation, which we believe is a worthwhile issue to address.

---

### Meta-Review · Area_Chair_HKkw · 2023-09-19

**Recommendation:** 3

**Metareview:**

This paper addresses the significant challenge of mitigating the distribution shift caused by using synthetic images during inference in multimodal machine translation (MMT). The authors propose a method that feeds both synthetic and authentic images to the MMT model, reducing the gap between their representations through Optimal Transport theory and ensuring output distribution consistency using KL divergence. The paper's contributions include the introduction of synthetic images during training, an optimal transport-based training objective, and a prediction consistency loss. The paper's contributions seem promising, but further exploration and addressing potential limitations may enhance this paper, some concerns regarding novelty and robustness have been raised.

Pros:

* Interesting Approach: The paper introduces an Interesting approach to addressing the dependence on authentic images in MMT. By leveraging principles like Optimal Transport and KL divergence, it proposes an Interesting technique to mitigate the distribution shift caused by synthetic images during inference.

* Positive Results: The paper provides a comprehensive evaluation against various baseline models and datasets, including the Multi30K dataset. This thorough evaluation enhances the validity of the approach and demonstrates its effectiveness.

* Clear Presentation: Reviewer 1 commends the clear writing and organization of the paper, making it easy to follow and understand.


Cons:

* Novelty is limited: Reviewer 1 raises concerns about the novelty of the proposed method, noting that the objectives and consistency concepts are not entirely new and have been used in other visual-text models.

* Potential Robustness problem: There are concerns about the robustness of the approach due to its heavy reliance on synthetic images during training. Reviewer 2 highlights potential sensitivity to the quality and diversity of synthetic images and the risk of overfitting to specific characteristics, limiting generalizability.

In summary, the paper offers an interesting approach to mitigate distribution shift in MMT caused by synthetic images during inference. It presents a thorough evaluation and demonstrates promising results. However, concerns regarding novelty and robustness have been raised.

---

### Decision · Program_Chairs · 2023-10-07

**Decision:**

Accept-Main

**Comment:**

This paper addresses the significant challenge of mitigating the distribution shift caused by using synthetic images during inference in multimodal machine translation (MMT). The authors propose a method that feeds both synthetic and authentic images to the MMT model, reducing the gap between their representations through Optimal Transport theory and ensuring output distribution consistency using KL divergence. The paper's contributions include the introduction of synthetic images during training, an optimal transport-based training objective, and a prediction consistency loss. The paper's contributions seem promising, but further exploration and addressing potential limitations may enhance this paper, some concerns regarding novelty and robustness have been raised.

Pros:

* Interesting Approach: The paper introduces an Interesting approach to addressing the dependence on authentic images in MMT. By leveraging principles like Optimal Transport and KL divergence, it proposes an Interesting technique to mitigate the distribution shift caused by synthetic images during inference.

* Positive Results: The paper provides a comprehensive evaluation against various baseline models and datasets, including the Multi30K dataset. This thorough evaluation enhances the validity of the approach and demonstrates its effectiveness.

* Clear Presentation: Reviewer 1 commends the clear writing and organization of the paper, making it easy to follow and understand.


Cons:

* Novelty is limited: Reviewer 1 raises concerns about the novelty of the proposed method, noting that the objectives and consistency concepts are not entirely new and have been used in other visual-text models.

* Potential Robustness problem: There are concerns about the robustness of the approach due to its heavy reliance on synthetic images during training. Reviewer 2 highlights potential sensitivity to the quality and diversity of synthetic images and the risk of overfitting to specific characteristics, limiting generalizability.

In summary, the paper offers an interesting approach to mitigate distribution shift in MMT caused by synthetic images during inference. It presents a thorough evaluation and demonstrates promising results. However, concerns regarding novelty and robustness have been raised.